# CALAMARI: Contact-Aware and Language conditioned spatial Action MApping for contact-RIch manipulation

**Youngsun Wi**[1]     **Mark Van der Merwe**[1]     **Andy Zeng**[2]     **Pete Florence**[2]     **Nima Fazeli**[1]

[1]Robotics Department, University of Michigan     [2]Google Deepmind

{yswi, nfz}@umich.edu     {andyzeng, peteflorence}@google.com

https://www.mmintlab.com/calamari

**Abstract:** Making contact with purpose is a central part of robot manipulation and remains essential for many household tasks – from sweeping dust into a dustpan, to wiping tables; from erasing whiteboards, to applying paint. In this work, we investigate learning language-conditioned, vision-based manipulation policies wherein the action representation is in fact, *contact itself* – predicting contact formations at which tools grasped by the robot should meet an observable surface. Our approach, Contact-Aware and Language conditioned spatial Action MApping for contact-RIch manipulation (CALAMARI), exhibits several advantages including (i) benefiting from existing visual-language models for pretrained spatial features, grounding instructions to behaviors, and for sim2real transfer; and (ii) factorizing perception and control over a natural boundary (i.e., contact) into two modules that synergize with each other, whereby action predictions can be aligned per pixel with image observations, and low-level controllers can optimize motion trajectories that maintain contact while avoiding penetration. Experiments show that CALAMARI outperforms existing state-of-the-art model architectures for a broad range of contact-rich tasks, and pushes new ground on embodiment-agnostic generalization to unseen objects with varying elasticity, geometry, and colors in both simulated and real-world settings.

**Keywords:** Contact-rich Manipulation, Visual-language guided policies

## 1 Introduction

Contact-rich manipulation is ubiquitous in our day-to-day lives, encompassing a broad range of tasks including sweeping dust into a dustpan, wiping tables, erasing a whiteboard, and applying paint with a brush. A key challenge in performing these tasks lies in controlling the interactions between tools and their environments. For instance, when sweeping, it is crucial to ensure continuous contact between the bristles and the surface while directing the collected dust towards the dustpan.

Language-conditioned representations and policies are a promising approach to addressing the challenges of contact-rich manipulation, particularly for domestic applications. For one, language is a powerful tool for creating abstractions that enable generalization for a wide variety of tasks and environments. Secondly, language will be among the most common methods to command robots such as when performing tasks in the home. Recent work has demonstrated how large pretrained visual-language models (VLMs), such as CLIP [1] and PaLM-E [2], enable zero-shot transfer of visual-semantic reasoning based on language prompts and well-structured visual and language embedding spaces [3, 4, 5, 6, 7, 8, 9, 10]. However, previous efforts have predominantly focused on rearrangement-based tasks and have not adequately addressed the reasoning involved in contact-rich manipulations, thereby limiting their applicability to tasks like wiping, sweeping, or scooping.

In this paper, we introduce a novel language-conditioned and contact-aware spatial action map representation that predicts *planar* contact affordances – contact formations at which tools grasped by

7th Conference on Robot Learning (CoRL 2023), Atlanta, USA.

(A)                  (B)

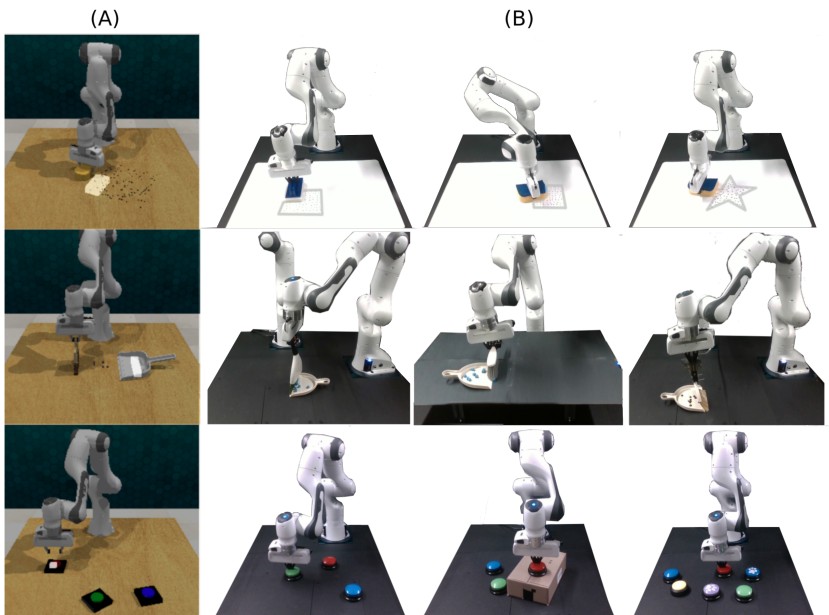

Figure 1: CALAMARI is a contact-aware and language-conditioned spatial-action mapping for contact-rich manipulation. We show that (A) wiping, sweeping, and pushing tasks, trained solely on simulations with a single task and prompt, can be (B) directly transferred to the real world and applied to new environments with unseen tools, robot setups, table elevations, and prompts.

the robot should meet an observable surface in order to perform a tabletop task. Our novel multi-modal spatial action maps are specifically for contact-rich manipulation where each pixel represents a binary indication of extrinsic contact between an object and the environment, and the entire map is implicitly linked to the tool pose and robot configuration. Notably, our extrinsic contact policies remains agnostic to intricacies of specific tools and physical robot platforms, unlocking possibilities for generalization to unseen objects with distinct elasticity, geometry, and colors, both in sim and the real world.

The key contributions of this paper are: 1) multi-modal extrinsic contact policy with novel spatial-action maps that outperforms SOTA language-guided manipulation methods for contact-rich tasks; 2) an MPPI controller algorithm compatible with the predicted contact goal and contact constraint; and 3) generalization to unseen objects with various elasticity, geometry, and colors in both simulation and the real world without requiring fine-tuning of the extrinsic contact policy.

## 2   Related Works

**Language Grounding for Manipulation:** The recent advancements in large language models (LLMs) [11, 12, 13] and visual-language models [1, 2] have enabled language-grounded manipulation. Numerous approaches [4, 9, 3, 14, 8, 15, 6, 16], have emerged and demonstrated remarkable multi-task performance, particularly for pick-and-place tasks. These end-to-end methods map directly from RGB or voxel observations to robot configurations. However, one drawback of end-to-end approaches is their reliance on extensive real-world data collection, which can span several weeks or months [15, 6]. To address this real-world data efficiency challenge, [9] achieved significant improvements by predicting only key frames and discretizing input and action spaces. Nevertheless, like end-to-end models, [9] faces the challenge of effectively handling novel tool manipulations, as the tool variations must be captured in the training demonstrations. We show that our approach CALAMARI can efficiently handle contact-rich tasks while reducing the burden on real-world data collection via zero-shot sim2real transfer.

**Planning with Extrinsic Contact:** Controlling extrinsic contacts for planning and manipulation has been an active area of research, with several notable contributions in recent years[17, 18, 19, 20, 21, 22, 23, 24]. Kim et al. [17] developed a method for simultaneously estimating and controlling extrinsic contacts of rigid objects using tactile signitures, while [18] focused on planning with extrinsic point contact for planar manipulation scenarios. Van der Merwe et al. [21, 22] demonstrated extrinsic contact detection for deforming tool manipulation, specifically scraping with spatulas, incorporating predefined contact goals and learned dynamics. Wi et al. [23, 24] presented a technique

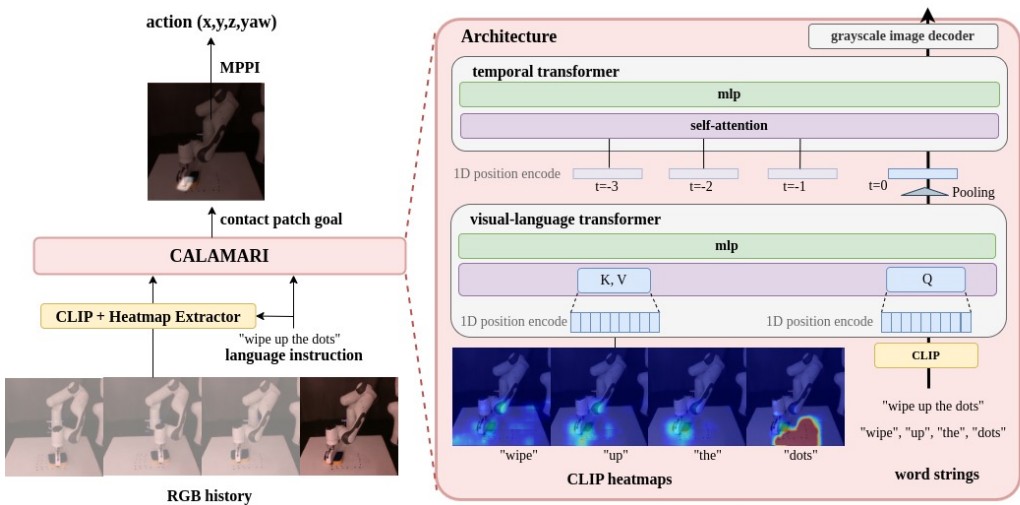

Figure 2: **Overview.** As shown in the left panel, our method utilizes the history of RGB and language instructions as inputs and predicts the contact patch goal as a binary mask from the input image frame. The three yellow blocks (e.g., 'CLIP') represent the pretrained models, which are not updated during training.

for predicting dense contact patches for compliant tools using learned tool dynamics. CALAMARI provides a framework for policies with contact goals, allowing for the seamless integration of the contact dynamics models [21, 22, 23] into a broader context of planning and manipulation.

## 3 Methodology

### 3.1 Problem Statement

Our objective is to learn a function $F$ that predicts the next desired contact patch goal, denoted as $\mathbf{C}_t^{goal}$, at time $t$. This function takes as input a sequence of RGB key frames and language instructions to compute contact goals: $F((\mathbf{I}_{t-w+1}, \boldsymbol{l}_0), \ldots, (\mathbf{I}_t, \boldsymbol{l}_0)) = \mathbf{C}_t^{goal}$, where $\mathbf{I}_t$ is an RGB image, $\boldsymbol{l}_0$ is a language instruction, and $w$ dictates the observation time window considered for making predictions. The output is the contact patch goal $\mathbf{C}_t^{goal} \in \mathbb{R}^{w \times h}$, a 2D binary mask in $\mathbf{I}_t$'s camera frame and is particularly well-suited to for contact-rich planar tasks [5, 25]. The contact patch $\mathbf{C}_t^{goal}$ can be de-projected to a point cloud by overlaying the predicted mask onto a depth map that excludes objects/tools ($\mathbf{D}_{nominal}$). The point cloud conversion enables the utilization of a model predictive controller to achieve the desired contact formation. In this work, data is provided in the form of demonstrations consisting of variable-length $T$ key frame trajectories, denoted by $\boldsymbol{\tau} = ((\mathbf{I}_0, \mathbf{C}_0, \boldsymbol{l}_0), \ldots, (\mathbf{I}_T, \mathbf{C}_T, \boldsymbol{l}_0))$. Here, $\mathbf{C}_t \in \mathbb{R}^{w \times h}$ is the contact patch between the object (e.g., grasped tool) and environment represented by a binary mask from the camera perspective. The contact patch from the demonstrations serves as the ground truth contact goal for the function $F$.

### 3.2 Behavior Cloning Tool-Environment Interactions

**Vision-Language Pre-Processing:** CALAMARI has two key vision-language pre-processing steps to convert raw observations into CLIP features. Firstly, we generate word-wise heatmaps that highlight the spatial locations in the RGB images corresponding to specific words in the language instruction. The Heatmaps, denoted as $\mathbf{H}_t = f(\mathbf{I}, \boldsymbol{l})$, are grayscale images with dimensions $w \times h$, obtained through the image-language relevancy extraction introduced in [26]. Using heatmaps instead of raw RGB input is particularly advantageous for sim2real and generalization to novel objects and environments, as long as similar heatmap distributions are present. This is because heatmaps provide abstract representation that is less sensitive to variations in visual appearance caused by factors like object colors and lighting conditions. The heatmap is subjected to further abstraction through a pretrained heatmap encoder [27], resulting in word-wise heatmap features. These features are organized based on the word sequence and serve as the query input for the Visual-language transformer, which will be discussed in detail in the subsequent section. Additional information about the selection of heatmap encoders can be found in Appendix-A.4.1.

Similarly, we convert the language prompts into embeddings using CLIP's language encoder. In this case, we combine both sentence and word embeddings to capture the contextual information

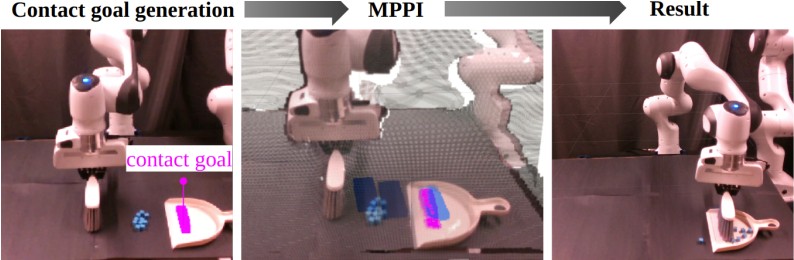

Figure 3: We generate the key contact goal via CALAMARI (magenta) and reach the contact goal via MPPI, which is linked with corresponding low-level actions. We visualize the tool's contact trajectory in blue until it reaches the contact goal. Once we have reached it, we generate a new contact goal until the task terminates.

conveyed by the language prompt while focusing on individual words. These embeddings are also arranged in accordance with the word order and serve as inputs for the subsequent transformer networks in the following section.

**Architecture:** The CALAMARI architecture (Fig. 2) consists of of two types of transformers, visual-language (v-l) transformer and temporal transformer. Drawing inspiration from the LAVA structure [14], our v-l transformer is responsible for encoding inputs into multi-modal features and our temporal transformer fuses latent observation over time to generate spatial-actions. In contrast to [14], the language query is comprised of a set of sentence and word embeddings, denoted as $\mathbf{Q} \in \mathbb{R}^{(l+1) \times d_{ft}}$. Here, $l$ represents the sentence length and $d_{ft}$ is the feature dimension. The keys and values, denoted as $\mathbf{K}$ and $\mathbf{V} \in \mathbb{R}^{l \times d_{ft}}$ respectively, are word-wise heatmap embeddings. The temporal transformer takes into account the history of latent observations from the v-l transformer, considering the $w$ most recent time stamps. Using the history of observation is important in contact-rich manipulation as it often involves occlusion caused by tools and the robot itself. The temporal transformer utilizes self-attention multi-modal features stacked with time, represented as $\mathbf{Q} = \mathbf{K} = \mathbf{V} \in \mathbb{R}^{w \times d_{ft}}$. The outputs of the temporal transformer are decoded using a grayscale image decoder, similar to the decoder architecture of the UNet model proposed by [28], without incorporating skipping layers.

**Training Loss:** The loss function is a standard supervised behavioral cloning loss similar to the prior works [6, 29, 30]. Specifically, we use L2 regression between predicted contact patch $\mathbf{C}_t^{goal}$ and the ground truth contact patch $\mathbf{C}_t^{gt}$ as the following: $\|\mathbf{C}_t^{goal} - \mathbf{C}_t^{gt}\|$.

### 3.3 MPPI controller and Contact Goals

We use the Model Predictive Path Integral (MPPI) [31] controller to plan a sequence of robot actions with corresponding contact patches to reach the desired contact goal $\mathbf{C}_t^{goal}$. The input to MPPI consists of the current pose of the end-effector and initial guess for the action trajectories. We define an action as the displacement in Cartesian SE(3) pose of the end effector and denote an action trajectory as $\boldsymbol{a} = (\boldsymbol{a}_0, \boldsymbol{a}_1, \ldots, \boldsymbol{a}_{w-1})$. Next, we define the controller cost:

$$\sum_{i=1}^{w} dist(\mathbf{C}_t^{goal} * \mathbf{D}_{nominal}, \mathbf{P}_{t+i}) + \lambda(1 - IoU(\mathbf{C}_t^{goal}, \mathbf{C}_{t+i}))$$

where $\mathbf{P}_{t+i}$ is the predicted contact pointcloud from applying the end-effector delta change in pose $\boldsymbol{a}_i$ to the object. We estimate $\mathbf{P}_{t+i}$ by transforming objects with known geometry to the world frame and identifying intersections with environment (Appendix A.5.4). $\mathbf{C}_{t+i}$ is a 2D projection of $\mathbf{P}_{t+i}$ to the camera frame. The first term of our cost function minimizes the mean Euclidean distance between the center of $\mathbf{P}_{t+i}$ and the contact goal center. To do so, we uses $\mathbf{C}_t^{goal} * \mathbf{D}_{nominal}$ to get the contact goal pointcloud by overlaying contact goal mask with the nominal depth map. The second term promotes matching the shape of future contact to the goal using Intersection over Union (IoU). We align the center of the prediction and the goal of contacts by subtracting the mean to focus on matching the contact shape (Appendix A.5.6).

Our MPPI has two contact constraints, implemented via penalty costs. The first constraint is to maintain contact via $\|C_t\| > 0$. This penalizes any actions that makes no contact. The second constraint is $\max_z(D_{nominal} - \mathbf{P_{t+i}}) > \epsilon_p$. This lower bounds the distance in $z$ axis from the environment to the transformed objects with epsilon $\epsilon_p > 0$. The entire control algorithm we use is described in Appendix Alg. 1.

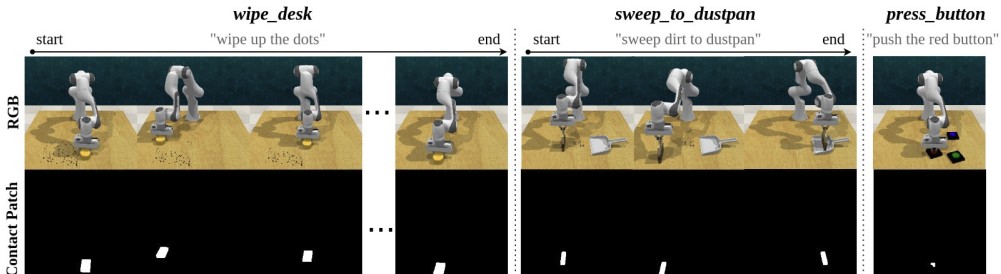

Figure 4: We visualize the dataset by displaying the language prompt alongside the RGB and contact patch sequences extracted from key contact frames in the demonstrations. The wiping task typically includes 7 to 12 key contact frames, sweeping involves 4 key contact frames, and the press button tasks have 1 contact patch.

## 4    Experiments and Result

**Datasets:** Both our model and baseline were trained using 100 demonstrations per task on CoppeliaSim [32, 33]. Each task involved manipulating a *single* object and a *single* language prompt, as described in Fig. 4. In Sec. 4.1, we investigate the generalization performance using various unseen objects and prompts. This paper focuses on three different types of contact-rich tasks: multi-step patch contact '*wipe_desk*', three-step patch contact '*sweep_to_dustpan*', and single-step point contact '*press_button*'. For the wiping task, we improvised our own closed-loop demonstration where the sponge moves towards the center of dust clusters computed from DBScan [34]. The other two tasks use open-loop demonstrations predefined in [33]. Each demonstration consists of a language prompt and a sequence of RGB observations along with the corresponding contact patches obtained from the key contact frames (Fig. 4 ). The key contact frames were identified when the robot reached the waypoints defined in the demonstrations. However, frames where the contact patch was the same as the previous waypoints were removed to eliminate redundancy. Ground truth contact patches are computed via CoppeliaSim's contact detection algorithm.

### 4.1    Simulation Results

The task performance of our model across the tasks is presented in Table 1, where the scores are averaged over 25 test episodes. Here, we utilized a 4 DoF action space $(x, y, z, yaw)$ to focus on planar manipulation, and we have included a full 6DoF manipulation result in Appendix A.5.3. We conducted three different objects/tools for evaluations: one being the training object and the other two being held out objects (Fig. 5). Our objective in directly transferring to these held out objects is twofold: 1) To demonstrate the robustness of our contact goal policy in effectively adapting to shifts in heatmap distribution from variations in object's structural/visual features. 2) To showcase the flexibility of our MPPI controller in accommodating previously unseen contact formations from unseen object geometries. Note that we employed new task-specific language prompts for the *push_button* task's heldout objects to accommodate color change.

**Evaluation Metrics:** The evaluation metrics are the task success rates ranging from 0% to 100%. We evaluate *wipe_desk* and *sweep_to_dustpan* tasks with the percentage of dust removal. For the *push_button* tasks, binary metrics were used, where 0% indicated failure and 100% represented success without partial credits.

*wipe_desk*: We evaluated the wiping performance after 20 contact goal generations, which is the point at which task performance plateaus (see Appendix Fig. 14). This task involves clearing one hundred dust particles with size 0.6x0.6x0.1cm that are randomly spawned within the bounding box of size 15cmx15cm. We show in Tab. 1 that CALAMARI achieves 98% of dust removal with the training object. To assess the performance with heldout tools, we conducted two tests. Test1 examined the ability of our MPPI controller to manipulate objects with 55% decreased contact patch area. Test2 evaluated the robustness of CALAMARI against out-of-distribution broom geometry with a long handle. As shown in Tab. 1, our method performance only decreased by 2% for test1 when handling smaller contact patches. Additionally, our goal generation exhibited robustness to heatmaps variations as test2 results are comparable to training object performance.

*sweep_to_dustpan*: We evaluated sweeping task with 3 contact goal generations as in the demonstration. *sweep_to_dustpan* task involves 5 dust particles with the size of 1x1x1cm. Details of the task environments are in Appendix A.2.1. Using training object, we achieved a 93% success rate in

| Method | wipe_desk | | | sweep_to_dustpan | | | push_button | | |
|---|---|---|---|---|---|---|---|---|---|
| | train | test1 | test2 | train | test1 | test2 | train | test1 | test2 |
| Ours | **98**% | 96% | **90**% | **93**% | **73**% | **84**% | **92**% | **60**% | **60**% |
| PerAct | 97% | **97**% | 9% | 86% | 0% | 0% | 63% | 0% | 24% |
| CLIPORT | 92% | 89% | 45% | 88% | 0% | 13% | 84% | 8% | 20% |

Table 1: RLBench success rate of each tasks in test cases using to train objects denoted as 'train' and heldout objects denoted as 'test1' and 'test2'

| wipe_desk | | | sweep_to_dustpan | | | push_button | | |
|---|---|---|---|---|---|---|---|---|
| training | test1 | test2 | training | test1 | test2 | training | test1 | test2 |
| 7x15x3cm | 4x12x3cm | 7x15x6cm | 1x10x15cm | 4x24x42cm | 5x34x8cm | 2.8cm | 2.8cm | 2.8cm |

Figure 5: Train and test objects in simulation of four different tasks. The heldout objects exhibit variations not only in color, size, and shape but also in structural attributes, such as the handle locations. The third row indicates object dimensions in centimeter either as $width \times depth \times height$ or as radius.

test cases for sweeping to a dustpan. We then transferred the pretrained model to two test objects: one with a longer handle, similar to the original, and another with a handle on the side. We noticed that larger visual discrepancies between the objects resulted in greater performance drops as test 2 shows worse performance than test1.

***push_button***: We altered the prompt "*push the {} button*" from the word "red" to "green" and "blue". This change resulted in varying CLIP heatmap intensities and word embedding inputs ,or the queries of our visual-language transformer. Nevertheless, our test results revealed that even with training based on a single word, our model achieved contact goal accuracy of less than 2.5cm in 60% of the two heldout cases. These findings emphasize the robustness and effectiveness of our approach in accurately pushing buttons of different colors.

**Baseline-PerAct:** In this section, we compare our methods with PerAct, a state-of-the-art language-conditioned manipulation study that also utilizes CLIP features. Details of implementation are described in Appendix A.3.1. Our method outperforms PerAct mostly across the three tasks (Tab. 1), both in training and testing with heldout objects. For the wiping task, PerACT shows significantly worse performance for test2 when compared to test1 and the training object. There are two main factors that explain the performance drop. First, PerAct is sensitive to larger changes in the transformations between the grasped position and the contact surface. We can observe this trend consistently across the sweeping tasks for the test objects as well. Second, PerAct is sensitive to variation in visual cues of the objects, as further supported by the sweep task. Our model is more robust as these changes have less impact on our contact planning than on the robot configuration. The results of push button task show that, while PerAct fails to detect buttons not in the training prompts, CALA-MARI leverages VLM's generalization ability for interacting with unseen prompts of different colored buttons.

**Baseline-CLIPORT:** CLIPORT is particularly suited for tabletop manipulation with 2D affordance prediction [5]. The CLIPORT baseline shares a number of important similarities to CALAMARI, including using an image action space and known tool geometry and pose. Details of implementation are described in Appendix A.3.2. Tab. 1 demonstrates that CALAMARI consistently outperforms the CLIPORT baseline. Wiping and sweeping results show that CLIPORT also struggles to generalize to unseen objects and tools with significant visual and geometrical variations. This is because CLIPORT encodes raw RGB-D without further abstraction unlike CALAMARI. Moreover, CLI-PORT also faces difficulty in generalizing to unseen prompts with color variations for the pushing task, similar to PerAct with significant drop in performance for these tasks.

## 4.2 Sim2Real

In this section, we directly transferred the pretrained model from simulation to the real world without fine-tuning. The quantitative analysis (Tab. 2) was conducted using 10 runs for each object with consistent resetting across all tasks. We utilized 2 Franka-emika robots and an Intel Realsense D435 camera for our real-world setup. The distance between the robot and camera between the real-world

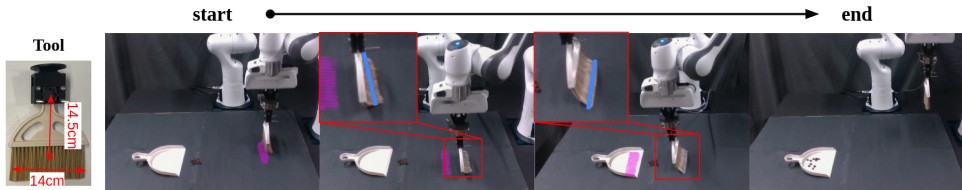

Figure 6: We demonstrate the ability of our model to generate goals for non-rigid tools. We repeat the sweeping task with a compliant tool, using a learned dynamics model to servo the contact of the tool with the tabletop. The predicted contact goal is visualized in magenta while the contact feature predictions from the dynamics model are overlaid in blue.

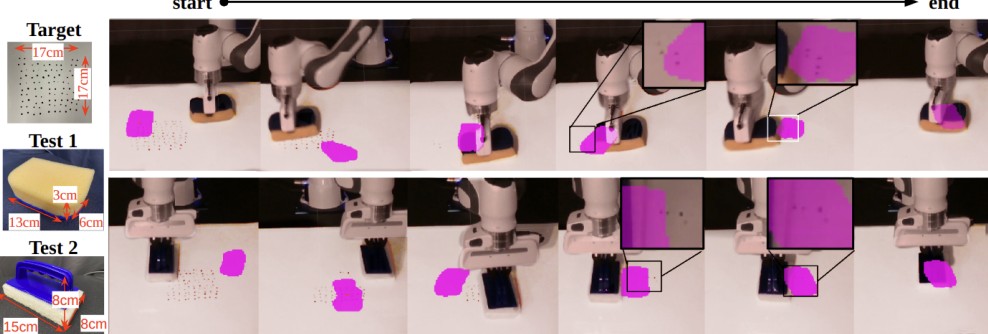

Figure 7: We visualized CALAMARI's contact patch goals in magenta. The first row represents the predictions when using the test1 object, while the second row corresponds to the test2 object. Our model can navigate back and forth until all the dots are erased using a closed loop policy. In real-world scenarios where the pressure exerted by a sponge on the board is not evenly distributed, this ability becomes particularly significant as the sponge may fail to erase certain dots even when it passes over them.

and simulation environments was [0.18m, 0.02m, -0.21m] in the x, y, and z directions. We found that the spatial-action map allows us to accurately predict contact patches in the camera frame regardless of the differences in camera positioning between simulation and real-world set-ups.

***wipe_desk*:** For the reset, we draw one hundred dots within a bounding box measuring $17cm$x$17cm$ with an black marker. We generated 20 contact goals per run and counted the number of erased particles for evaluation. Fig. 7 shows our contact patch goals, where we visualize the first and last three frames when contacts goals are generated. We also conducted experiment to erase different distributions of dots with the test1 object in Appendix Fig. 15.

***sweep_to_dustpan*:** We performed resets by arranging the dustpan in 10 different configurations within a bounding box measuring $9cm$x$14cm$. Each configuration included 10 $1cm^3$ cubes placed in front of the dustpan. For evaluation, we counted the number of cubes successfully swept into the dustpan. Fig. 8 visualizes our contact goals, which directed the broom to align with the dustpan and sweep towards it. Interestingly, our real-world results show a dust sweeping rates of 91-92% from test1 and test2 brooms, which surpassed performance in simulation with unseen tools.

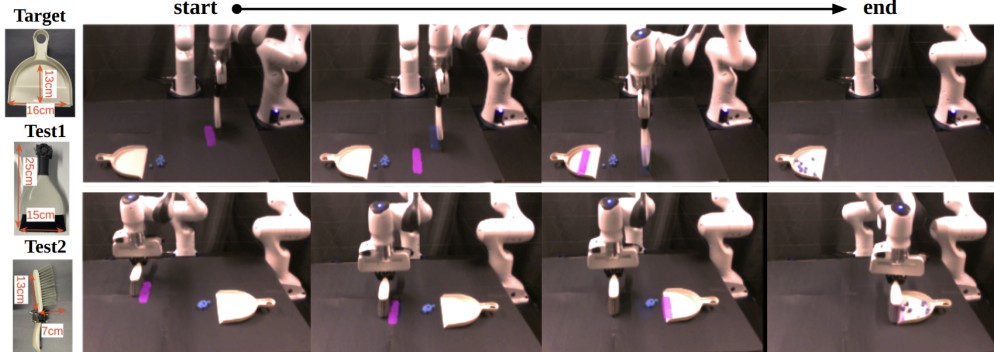

Figure 8: The real-world sweeping results were obtained using two brooms with small width margins (1-3 cm) with the target dustpan. Left panel shows three contact goal generations, visualized in magenta. The results in the last column demonstrate that our policy exhibits excellent zero-shot transfer to the real-world, even with unseen tool geometries. Furthermore, we demonstrate that our method is agnostic to the arrangement of the robot, as the same policy was applied to both the right and left arm.

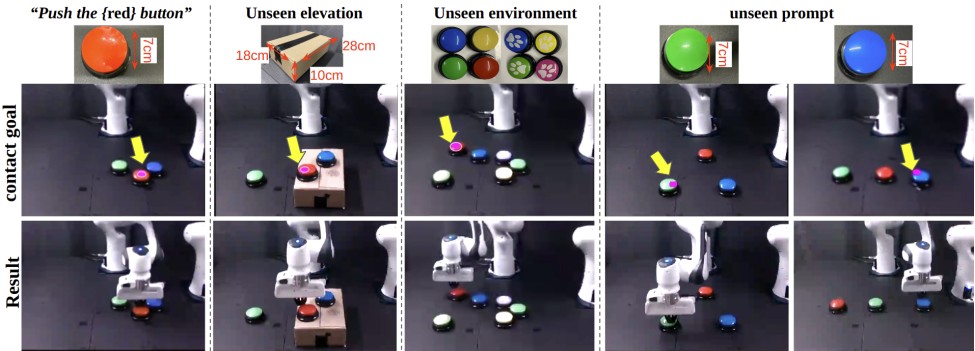

Figure 9: Our model, trained in CoppeliaSim on the prompt "press the red button," can be directly transferred to the real world without fine-tuning (first column). Our spatial action space is not limited to the tabletop (second column) and is robust to visual distribution (second and third column). It also handles unseen prompts like "press the green/blue button" (last two columns). The contact goal is visualized in magenta in the first row, while the second row shows the actual execution results

**push_button:** We positioned three buttons (red, green, blue) in 10 predefined line and triangular arrangements on the desk. For evaluation, we assessed whether the end-effector successfully made contact with the target button. The real-world outcomes aligned with the simulation results for the red and green buttons (training and test1), but pressing the blue button exhibited a performance discrepancy of 20%. Fig. 9 demonstrates the adaptability of our pretrained model to diverse, unseen setups involving variations in elevation and the number of buttons in the scene.

| | | |
|---|---|---|
| *wipe_desk* | test1 | 91% |
| | test2 | 98% |
| *sweep_to_dustpan* | test1 | 91% |
| | test2 | 92% |
| | comp. | 85% |
| *push_button* | red | 90% |
| | green | 60% |
| | blue | 40% |

Table 2: Direct transfer to real-world using test objects described in Fig. 7, 8, 9. 'comp' utilizes learned compliant tool dynamics as in Sec. 4.3.

## 4.3 Compliant Tool Manipulation

In this section, we demonstrate our model performance on a real world, compliant tool manipulation task. By decoupling goal generation and dynamics, our method can generate valid contact goals for tasks using compliant tools, so long as dynamics of the tool are available. We execute the *sweep_to_dustpan* task with a compliant brush (compare tool deformation in Fig. 6 to Fig. 8). Note that the goal generation is learned on rigid tools in CoppeliaSim and transferred without finetuning to a deformable tool in the real-world. We replace rigid dynamics with a learned contact feature dynamics to estimate $\mathbf{P}_{t+i}^{dyn}$ [21]. Full details of the contact feature dynamics can be found in Appendix A.2.5. Quantitative results are shown in Tab. 2 and a qualitative sweep is shown in Fig. 6. Our method can effectively predict contact goals for a deforming tool, yielding 82% performance.

## 5 Limitations

Our approach offers versatility in manipulating objects on various planar manipulation scenarios, including elevated and potentially inclined planes. However, our ability to predict contact is limited to a 2D binary contact patch, therefore, it is challenging to directly apply our method for more intricate contact-rich manipulation scenarios like screwing bulbs or peg insertions. Moreover, we assume the region of interest (e.g., areas to wipe, sweep, or push) is already within the camera's field of view. Lastly, our approach lacks support for discontinuous contact. As a future work to enable CALAMARI to effectively address more complex contact-rich scenarios, we suggest an extension to 3D contact mask prediction, potentially leveraging state-of-the-art surface reconstruction techniques (e.g., [26] ). Additionally, we suggest integrating binary contact prediction with a contact mask to inform the controller to switch between free space motion and in-contact mode.

**Acknowledgments**

This work was supported by the National Science Foundation (NSF) grant NRI-2220876. Any opinions, findings, and conclusions or recommendations expressed in this material are those of the authors and do not necessarily reflect the views of the National Science Foundation.

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

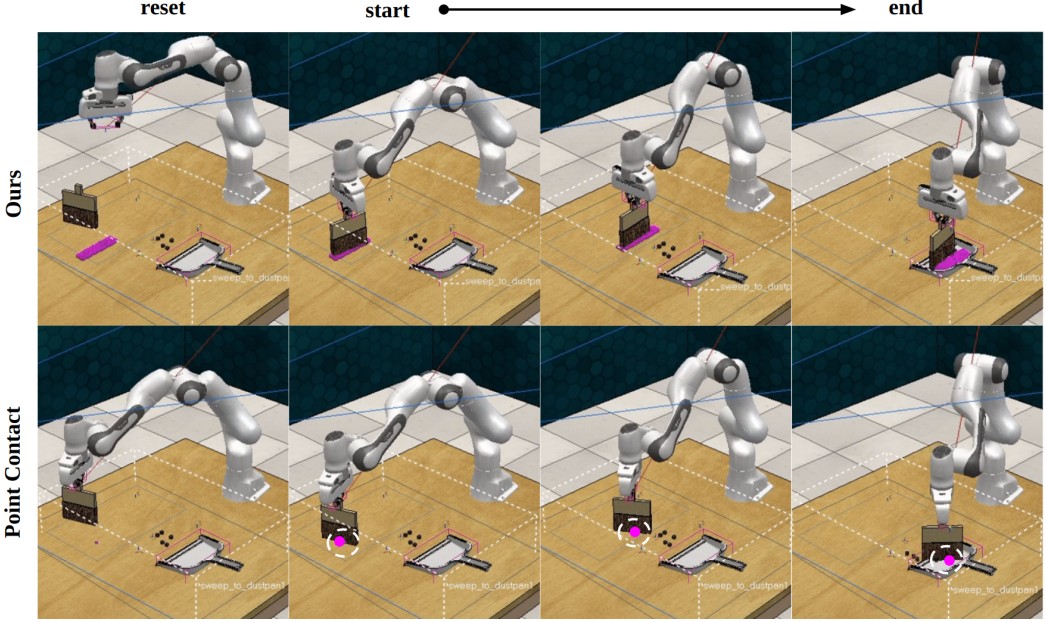

Figure 10: Using contact goals as points instead of patches results in a loss of control over the contact orientation. As demonstrated in this figure, CALAMARI successfully reorients the broom to maximize dirt sweeping. The magenta color indicates contact goal in all figures.

# A  Appendix

## A.1  Ablation Studies

### A.1.1  Contact Patch Verses Point Predictions

Predicting contact *patches* is crucial for precise contact-rich manipulation. For instance, optimizing contact patch orientation can greatly increase efficiency in sweeping up larger amount of dust simultaneously. Our proposed model-predictive control approach tracks desired contact patches, in particular matching contact patch size and orientation. To demonstrate the importance of contact patch control, we present our validation in the following ablation section and Fig.10 where the initial broom orientation is set to $\pi/4$ rad to the world-frame. Here, we extracted the center of the contact patch goal and executed our MPPI algorithm without the Intersection over Union (IoU) cost. Our results show that using the point contact goal results in a 14 % success rate, which is significantly lower than CALAMARI's performance with contact patch goal and the same orientation offset (93 %).

### A.1.2  Temporal Transformer

To evaluate the impact of CALAMARI's temporal transformer, we conducted a comparison between the wiping task performance with and without the temporal transformer (Tab. 3. We chose the wiping task because it is characterized by the longest planning horizon among our tasks and exhibits the most significant scene occlusions due to the robot arm and tool. In the 'without temporal' experiment, we directly used the latent state from the visual-language transformer as an input to the grayscale contact goal decoder. We kept the hyperparameters fixed across trials. The training loss curve is shown in Fig. 11. We note that the task success rate using train object without temporal transformer was 93%, which is 4% lower than that of our proposed architecture.

## A.2  Results Details

### A.2.1  CoppeliaSim Sweep Task

| wipe _desk | train | test1 | test2 |
| --- | --- | --- | --- |
| CALAMARI | 97 % | 93% | 78% |
| without temporal | 93 % | 86% | 74% |

Table 3: Performance analysis with and without temporal transformer.

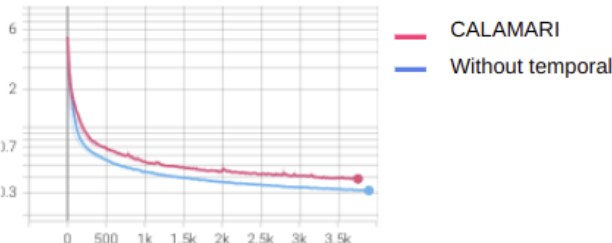

Figure 11: Training loss curve of CALAMARI with and without temporal transformer.

Among our tasks, *sweep_to_dustpan* involves two different objects in the scene: broom and dustpan. To assess the robustness of CALAMARI and the baselines towards unseen scenarios, both objects were modified, as illustrated in Fig. 12. Each broom was created from scratch, designed after real-world brooms commonly found in retail stores. As for the dustpan, we employed the same mesh used for training, but altered its color to match that of the broom. Additionally, we introduced geometric variance by scaling the dimensions along the x, y, and z directions. We note the dustpan dimension for training was $25cm$x$30cm$x$7cm$, while for Test1, the dimension was $20cm$x$35cm$x$17cm$, and for Test2, the dimension was $20cm$x$25cm$x$3cm$.

### A.2.2 Real-world Setup Details

Fig. 13 shows our experimental setup using a single accessible vision sensor from the front view. We note that only one arm was used for manipulation. We found that RGB image noise could adversely influence the scale of the heatmap signal, which in turn affect goal generation of our method on the *sweep_to_dustpan* task. This is due to the heatmap's sensitivity to high-frequency RGB noise, not present in simulation, resulting in a mild divergence in subsequent contact goal predictions for nearly identical inputs. As such, for the *sweep_to_dustpan* task, and compliant variant, we repeated each trial twice and reported the better performing result.

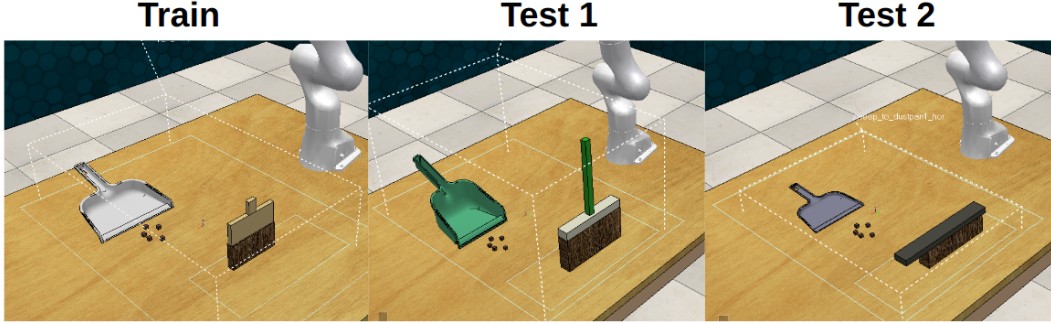

Figure 12: For the sweeping, we not only used unseen brooms, but also augmented the dustpan to different colors and dimensions.

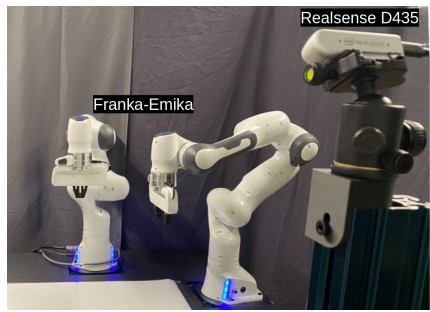

Figure 13: We used 2 Franka-emika robots and an Intel Realsense D435 for real-world setup.

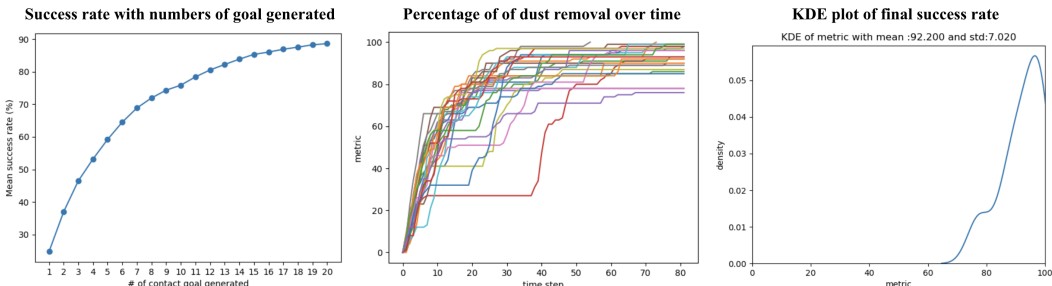

Figure 14: Details on wiping performance with training object over the number of goal generated (first) and time (second). Second plot visualizes each 25 tests' dust removal rate over time. Finally we show the distribution of success rate after 20 goals.

### A.2.3 Real-world Wipe Desk

Fig. 15 shows generalization of our wiping task to unseen dot arrangements. This results in different heatmap distributions. As in the main experiment, we run the wiping until it generates 20 contact goals.

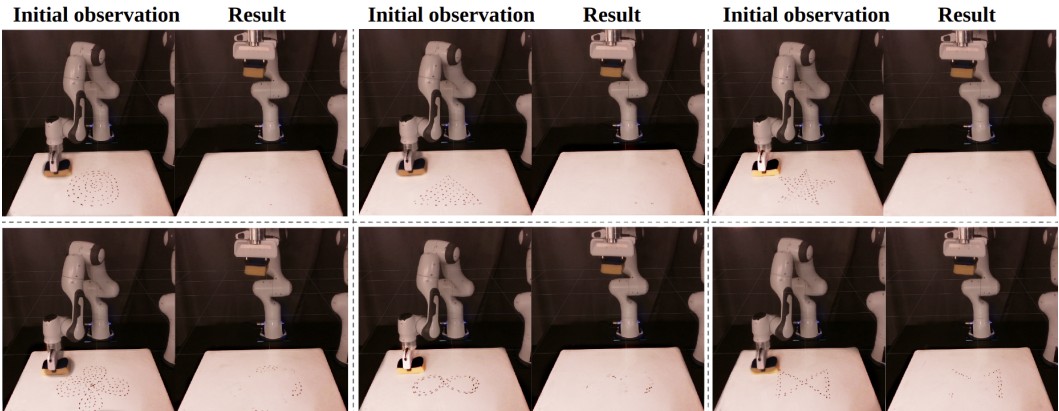

Figure 15: We examined the wiping results using the test1 object on different dot arrangements, leading to an unseen heatmap distribution for the model, which was originally trained only on square-shaped dot clusters. From the top left to the bottom right, we conducted tests on circle, triangle, star, quatrefoil, infinite, and hourglass shapes. Our observations show the first four shapes were nearly entirely erased after the wiping. In contrast, we observed a disparity in performance with the last two shapes, characterized by unfilled holes in their centers, leaving behind 10% and 54% of the dots, respectively.

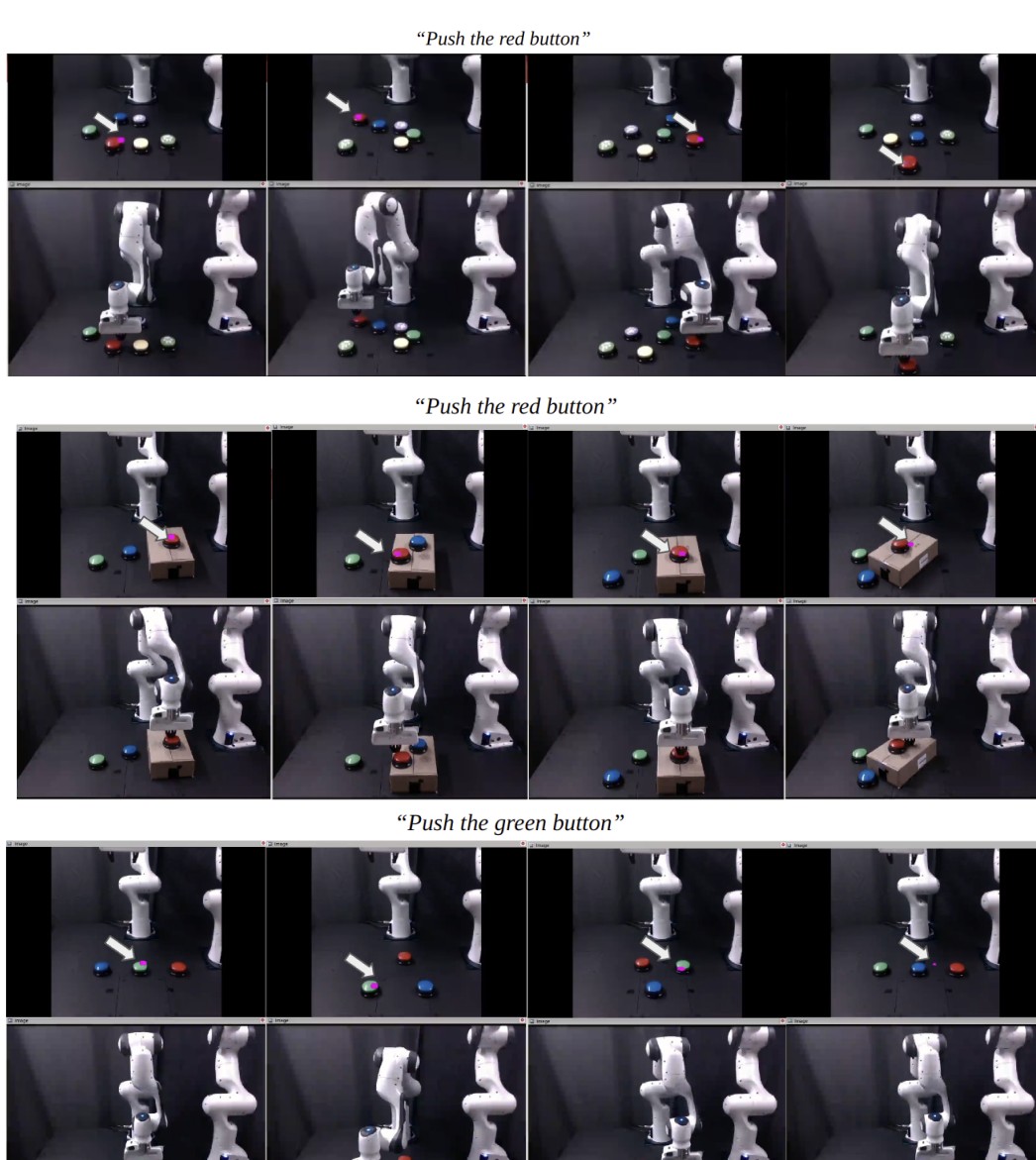

Figure 16: First row indicates generated contact goal indicated in magenta pixels. The second row indicates robot execution results.

### A.2.4 Real-world Push Button

Fig. 16 shows more example results on button pushing task with unseen numbers of buttons in the scene, unseen table elevation, and unseen prompts.

### A.2.5 Compliant Sweep Task

Our method decouples contact goal generation from the low-level controller responsible for realizing the contact goals. To control contact between a deforming tool and the tabletop, we use the contact feature dynamics model from Van der Merwe et al. [21]. The model predicts contact geometries (represented as lines in 3D) given candidate actions, conditioned on point cloud and wrench observations. The contact point cloud $\mathbf{P}_{t+i}$ is obtained by sampling evenly between the end points of the predicted contact line.

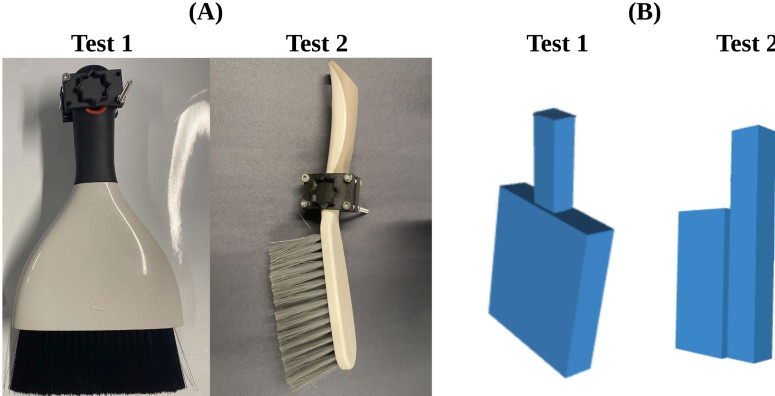

Figure 17: (B) depicts the rigid object geometries utilized in our experiment, corresponding to the actual objects shown in (A). While the object models do not encompass the intricate details of the object geometry, our research demonstrates that relying solely on the coarse dimensions of the geometry – such as the object's collision model, commonly employed for collision avoidance during robot planning – proved sufficient for successfully executing our task.

We train the model by performing randomized actions and label contact lines using a heuristic on the observed point clouds. Specifically, we threshold for points near the surface, then fit a line to the resulting points projected onto the tabletop. Point cloud observations are obtained by a Photoneo Phoxi 3D scanner (L) and a Photoneo MotionCam-3D Color (M+) scanner. Wrench observations are obtained by an ATI gamma force torque sensor, attached between the end effector and compliant tool. The dynamics model is trained on 3236 sampled transitions.

### A.2.6  Grasped Tool Geometry

For the object geometries, we crafted their meshes via *trimesh* Python package utilizing rough object dimensions as shown in Figure 17(B). Constructing meshes required less than 10 minutes to create each for the objects we employed. While we haven't specifically explored automating this process in our current work, it is plausible that SOTA 3D shape completion techniques from partial object point cloud measurements can be used to generate meshes.

### A.2.7  Task Performance Robustification

We achieved robustification of contact goal generations by applying robot workspace mask to the RGB images using a known transformation from the camera to the robot and by employing heatmap augmentations. These heatmap augmentations include flipping and translation( both horizontal and vertical) ranging from 10 to 30 pixels. Furthermore, we introduced four different levels of Gaussian noise to the RGB before generating the heatmap, enhancing the robustness of our method against high-frequency noise in real-world RGB images.

### A.3  Baseline Details

### A.3.1  PerAct

PerAct employs voxel inputs from a multiple calibrated RGB-D camera setup, while our method relies solely on a single front camera view. In contrast, our MPPI controller requires additional information about the model's geometry and object pose, whereas the baseline does not require such object-related details. For this experiment, PerAct was trained from scratch with a 512 latent dimension. To align the baseline with our experimental setup, we trained PerAct with a single-task using the same dataset. During testing, we reduced the action space to (x, y, z, yaw) by providing

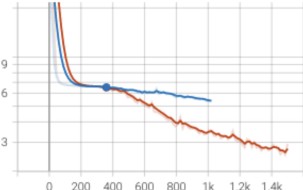

Figure 18: Blue is a training loss curve when training image encoder from scratch and the red is the trainind loss curve when training with frozen pretrained ResNet18. Other than the encoder, we used the same training settings. The dataset used for this experiment is 50 demonstrations of *press_button* dataset.

ground truth values for the other actions (pitch, roll, gripper state, and collision prediction) as well as ground truth grasping.

### A.3.2 CLIPORT

CLIPORT's inputs consist of a single RGB-D and language instruction, whereas the outputs are three affordances for picking, placing, and a discrete end-effector angle for placing. To elaborate, CLIPORT outputs (u,v) for picking and (u,v,yaw) for placing, where u and v denote pixel coordinates in the tabletop view. These table-top pixel coordinates are then projected into the world frame (x,y,z) using a known transformation. Subsequently, we employ the known tool geometry and pose to transform the target point (x,y,z) into the robot end effector frame, ensuring that the bottom center of the tool reaches the target point with the desired wrist rotation (yaw). We provide ground truth z corresponding to the table height.

### A.4 Input Processing Details

### A.4.1 Heatmap Encoder

We implemented two gray-scale image encoder. The first was based on pytorch's ResNet implementation by changing the first encoder's input dimension as 1 instead of 3. We used 4 residual blocks and 2 convolution layers for each residual blocks following the original pytorch implementation. We trained this encoder from scratch along with other modules of CALAMARI. The second was the pretrained ResNet18 [27] trained on ImageNet. The network is for RGB image, such that we repeated the grayscale heatmap inputs for three times and stacked in depth to match the desired the input dimension. We note that the pretrained image encoder's parameters are frozen without fining tuning. Our experiment shows that the second encoder gives faster convergence and lower training loss as shown in Fig. 18.

### A.5 Controller Details

### A.5.1 Control Pipeline

Alg. 1 explains our hierarchical controller in the context of other components of CALAMARI. When given the contact patch goal, we repeat a control process consisting of the MPPI (Alg. 1 line 8) and the impedance controller (Alg. 1 line 9). The control loop continues until the grasped objects achieves the contact goal, measured as *cost* using the same MPPI cost function (Alg. 1 line 5). If the current pose of the object resulting from the impedance controller has a smaller cost than the threshold $\delta$, a new contact goal is generated via *generate_contact_goal* using visual-language observation(Alg. 1 line 5). Task is finished when the task objective is achieved (Alg. 1 line 12); for example, when all the dust is swept, when all the dots are erased, and when the button has been pushed correctly. Tasks are also finished when the number of contact goals exceed the number of goal threshold (Alg. 1 line 14), where the contact goal threshold ($g_{thres}$) is set to 20 for wiping, 3 for sweeping, and 1 for pushing.

Next, we provide additional details about our MPPI controller, building upon the description provided in Sec. 3.3. The role of MPPI is to compute a sequence of robot actions that result in the

**Algorithm 1**

---

1: $t \leftarrow 0$
2: $g \leftarrow 0$
3: $complete \leftarrow False$
4: **while** not $complete$ **do**
5:     **if** $cost(s_t) \leq \delta$ **then** $\mathbf{C}^{goal} = generate\_contact\_goal(obs_t)$
6:         $g \leftarrow g + 1$
7:     **end if**
8:     $a_t \leftarrow mppi(s_t, \mathbf{C}^{goal})$
9:     $env.step(a_t)$
10:     $t \leftarrow t + 1$
11:     **if** $task.get\_completed$ **then**
12:         $complete \leftarrow True$
13:     **end if**
14:     **if** $g > g_{thres}$ **then**
15:         $complete \leftarrow True$
16:     **end if**
17: **end while**

---

desired contact goals given by our representation, Fig. 3. Here, we define an action as the change in Cartesian SE(3) pose of the end effector and denote an action trajectory as $\boldsymbol{a} = (\boldsymbol{a}_0, \boldsymbol{a}_1, \ldots, \boldsymbol{a}_{w-1})$. The input to MPPI consists of the current pose of the end-effector and initial guess for the action trajectories. Given the cost function described in Section 3.3, the output of MPPI is the action sequence with the lowest cost. To predict contact locations using action samples, given a goal at each time step, we predict contact patches of a tool given sampled end-effector actions. For rigid object contact estimation, we apply the end-effector Cartesian change in position to the grasped tool (assuming a fixed grasp) and compute the intersecting geometry with the environment resulting in the contact patch. This is achieved by comparing the transformed tool mesh/pointcloud with the $\mathbf{D}_{nominal}$. For the compliant tool, we use the Extrinsic Contact Servoing approach [21] which directly yields the contact lines, utilizing wrench measurements and a partial point cloud given the action to be executed. One may substitute of models including [23].

### A.5.2 Controller Parameters

For the mppi, the actions were sampled from $\mathcal{N}(\mathbf{0}, \sigma)$ and clipped using the action bound. Our MPPI framework is based on the external repository following the prior works [21] and we set the parameters for the mppi as follows: action_high $=[x, y, z, r, p, y]$=[0.04, 0.04, 0.001, 0., 0., 0.3], action_low=$[x, y, z, r, p, y]$=[-0.04, -0.04, -0.001, 0., 0., -0.3], num_samples=1000, nx=6, lambda= 0.000001, horizon=1, and

$$
\sigma = \begin{bmatrix}
0.01 & 0. & 0. & 0. & 0. & 0. \\
0. & 0.01 & 0. & 0. & 0. & 0. \\
0. & 0. & 0.001 & 0. & 0. & 0. \\
0. & 0. & 0. & 0.0005 & 0. & 0. \\
0. & 0. & 0. & 0. & 0.0005 & 0. \\
0. & 0. & 0. & 0. & 0. & 0.01
\end{bmatrix}
$$

.

### A.5.3 6-DoF Manipulation

In this section, we present the results of the full 6DoF manipulation by expanding the action bounds of $r$ and $p$, as well as the noise sigma. Now, all rotation components share the same action bounds: action_high = $[x, y, z, r, p, y]$ = [0.04, 0.04, 0.001, 0.3, 0.3, 0.3], and action_low = $[x, y, z, r, p, y]$ = [-0.04, -0.04, -0.001, -0.3, -0.3, -0.3]. Other MPPI settings, such as the number of samples, lambda,

| $(\sigma_{\mathbf{r}}, \sigma_{\mathbf{y}})$ | wipe_desk | sweep_to_dustpan |
|---|---|---|
| (0.01, 0.01) | 95 % | 92 % |
| (0.05, 0.05) | 94 % | 87% |
| (0.1, 0.1) | 91 % | 84% |

Table 4:

6-DoF manipulation experiments with different roll and pitch action sampling parameters.

horizon, and more, remain unchanged. Consequently, we update the action noise sigma as:

$$\sigma = \begin{bmatrix} 0.01 & 0. & 0. & 0. & 0. & 0. \\ 0. & 0.01 & 0. & 0. & 0. & 0. \\ 0. & 0. & 0.001 & 0. & 0. & 0. \\ 0. & 0. & 0. & \sigma_r & 0. & 0. \\ 0. & 0. & 0. & 0. & \sigma_p & 0. \\ 0. & 0. & 0. & 0. & 0. & 0.01 \end{bmatrix}$$

Here, $\sigma_r$ and $\sigma_p$ are scalar values associated with the standard distribution of roll and pitch actions, respectively. In Tab. 4, we evaluate different combinations of $(\sigma_r, \sigma_p)$ in two of our more intricate control tasks, using the "train object" in Coppeliasim. The final row, which utilizes $(\sigma_{\mathbf{r}}, \sigma_{\mathbf{y}})$ = (0.1, 0.1), representing a scenario where all rotation components share the same action sampling distribution. The Tab. 4 shows the task performance may decrease as we increase the action noise. Keeping the number of samples constant while increasing action noise for MPPI results in sampling deficiencies, which lead to suboptimal control result. Finally, our robot's Cartesian impedance controller stiffness parameters was set to [3000.0, 3000.0, 3000.0, 300.0, 300.0, 300.0] for rigid tool manipulation and [2000.0, 2000.0, 1000.0, 100.0, 200.0, 200.0] for complaint tool manipulation.

### A.5.4 Contact Patch Prediction

To obtain the intersections between the object's point cloud and the environment, we first extract the lowest 10% of the point cloud, which serves as the contact candidates. Using contact candidates enhances calculation efficiency. If the z-coordinate of the contact candidate is lower than or equal to that of the closest point in the environment in Manhattan distance on the x and y axes, we consider the candidate pixel to be in contact.

### A.5.5 Environment Geometry

We define the environment as a point cloud or the depth map without tools and non-collidable objects. In the CoppeliaSim simulation, we utilized *set_model_renderable(False)* for the tools and non-collidable objects in the scene during task initialization. In the real world, we set aside the tools from the camera's angle at the task initialization and capture a depth map to obtain the environment geometry.

### A.5.6 Intersection over Union

Before calculating the IoU, we align the centers of contact patch predictions with the contact goal mask to ensure that the IoU metric focuses purely on shape matching. This alignment is achieved by subtracting the mean pixel values of each 2D mask. We found that using IoU metrics without this center offset can lead to undesirable yaw movements of the object. This is because yaw motions can lead to higher IoU scores by creating larger overlaps between the tool's actual contact and the contact goal, especially when the tool can only partially reach the goal with the action. Consequently, IoU scores without this alignment do not accurately reflect the deviations from the desired contact's shape and orientation.

