# OpenReview forum: "CALAMARI: Contact-Aware and Language conditioned spatial Action MApping for contact-RIch manipulation"
_robot-learning.org/CoRL/2023/Conference — CoRL 2023 Poster_

### Official Review · Reviewer_eq6y · 2023-07-12

**Confidence:** 4
**Originality:** Very Good
**Technical Quality:** Good
**Clarity Of Presentation:** Excellent
**Impact:** 4

**Recommendation:**

Weak Accept: I recommend accepting the paper, but will not argue for my recommendation if the majority of other reviewers have a different opinion.

**Review:**

Strengths:
+ Overall, the idea of fine-tuning large vision-language models to predict contact heatmaps, and then using model-based controllers to achieve those contract interactions, is novel and interesting. Vision-language models do not reason about actions, and end-to-end behavior-cloning agents fail to generalize beyond the training distribution. Prior works like CLIPort have attempted to reconcile these two aspects, but they are often limited to 2D pick-and-place tasks. CALAMARI cleverly adapts a vision-language model to predict contact interactions.
+ CALAMARI achieves compelling empirical results both in simulation and real-world environments. CALAMARI outperforms PerAct (a state-of-the-art language-conditioned manipulation agent) in a reproducible setup. Specifically, CALAMARI generalizes better to unseen objects during test time.
+ Generally, the paper is well-written and easy to understand. The figures are informative. The result tables are easy to parse. The supplementary material provides additional details for reproducibility.

Weaknesses:
- The diversity of evaluation tasks is very limited. Is there a particular reason why there are only 3 task categories: wiping, sweeping, and push button? It’s understandably difficult to conduct large-scale evaluations in real-world settings, but CALAMARI seems applicable to a lot more RLBench tasks. More diverse tasks might also provide better insights into the capabilities and limitations of CALAMARI’s approach.
- It seems like the MPPI controller makes additional assumptions about object geometry and pose (Line 192). Does the MPPI controller need a perfect 3D reconstruction of the object? If so, how were these reconstructions provided in the real-robot setup? Assuming good geometry and pose somewhat limits the generality of predicting contact patches in RGB observations.
- The experiments are missing some ablation studies. For instance, how important is providing a history of RGB observations, and how sensitive is the model to history length? How important are the positional encodings? These are some suggestions, but it would be great to see which design choices were crucial for CALAMARI’s success.
- For certain tasks, what if the contact region is occluded in the RGB input? For example, when wiping a surface that is facing away from the camera, it would be difficult to predict contact patches without any visual feedback. If this is a limitation, it should be discussed in the limitations section.
- Minor: It would be nice to have a supplementary video showcasing the system. The agility and robustness of contact-rich systems are much easier to understand with videos and demos.
- Minor:  Section 5 “Limitation” -> “Limitations”. Also, it would be nice to have a conclusion section to wrap-up with main takeaways from the paper. But this section was understandably cut due to space limitations.


**Quality Of The Limitations Section:**

Limitations are addressed clearly

**Questions For Rebuttal:**

Summarizing the questions from above:
- Have you tried evaluating on more simulated tasks?
- Does the MPPI controller need a perfect 3D reconstruction and an accurate pose of the object?
- How important are the individual components of CALAMARI?
- What if the true contact regions are occluded by objects in the scene?


**Robotics Focus:**

Sufficient demonstration on hardware

**Summary Of Paper:**

This paper presents CALAMARI, a language-conditioned manipulation agent that models actions as contact interactions. CALAMARI encodes a history of RGB observations, and predicts contact heatmaps in image-space. These heatmaps are initialized with CLIP saliency, and fine-tuned with in-domain data. These contact heatmaps are then used with an MPPI controller to optimize 4-DoF actions that achieve the desired contact interactions. Simulation experiments are set in Coppeliasim, where CALAMARI is benchmarked against PerAct. The results show that CALAMARI significantly outperforms PerAct, especially with unseen test objects. Additional experiments show good sim-to-real transfer and some preliminary capabilities with compliant tool manipulation.


**Summary Of Recommendation:**

CALAMARI is a novel framework that predicts contact heatmaps, and then uses an MPPI controller to achieve those contact interactions. CALAMARI achieves compelling results both in simulated and real-world manipulation tasks. The experiments could be improved by evaluating on more diverse tasks and conducting ablation studies, but overall, CALAMARI is a good addition to the venue.

**Post-Rebuttal**
The authors' response is very helpful in clarifying initial doubts and questions. The reviewer thanks the authors for conducting additional experiments and adding more details to the paper. Despite this, the reviewer is still concerned about the limited diversity of tasks (which could have been scaled up in simulation), and some strong assumptions about object geometry for the MPPI controller. Based on these two key issues, I am keeping my original score of Weak Accept.

---

### Official Review · Reviewer_hB1X · 2023-07-17

**Confidence:** 4
**Originality:** Good
**Technical Quality:** Good
**Clarity Of Presentation:** Very Good
**Impact:** 3

**Recommendation:**

Weak Accept: I recommend accepting the paper, but will not argue for my recommendation if the majority of other reviewers have a different opinion.

**Review:**

This paper presents an interesting robot learning system that is suitable for contact-rich tasks and demonstrates its superior performance over baselines in simulation and in the real world with zero-shot sim2real transfer. The model is able to generalize to novel objects (tools), robot setups, lighting conditions, table elevations, and prompts.

This work has a few noteworthy strengths. First of all, the algorithmic design seems intuitive but pretty effective. By leveraging the CLIP heatmaps, the model is robust to variations in object shape/texture/lighting conditions/sim2real gap, etc. The two layers of transformers to fuse multimodal features and temporal features seem pretty effective as well. The model also seems to be fairly reactive and can accommodate different types of elasticity of the tool (repetitively attempt to accomplish the task if not done yet).

Secondly, because of the algorithmic design, the model is able to achieve zero-shot sim2real transfer with minimal performance drop. One of the main benefits of this is that only demonstrations in the simulation need to be (programmatically) collected. The experimental evaluation is well designed and three evaluation tasks are representative contact-rich tasks.

Thirdly, the paper is relatively well-written. Different components of the system are well explained and the figures are pretty illustrative.

This work, however, is not without limitations. One of the main limitations is that low-level MPPI controller requires ground truth object geometry and pose during inference time. This is okay and the authors made it very clear this is the case in the paper. However, this made it difficult or basically unfair to compare CALAMARI with PerAct, which is the only baseline that the proposed method is compared to. Also, the three tasks being evaluated, although representative, don’t seem to be too challenging. In fact, it’s a bit unclear if a contact patch area prediction is needed, as opposed to a contact “point” prediction. It would be interesting to evaluate the model on a more challenging task of “apply paint” (which is mentioned in the intro of the paper), which requires the robot to draw a specific shape/alphabet on the whiteboard. In this case, I would imagine a precise contact patch area prediction is needed (a point estimate is too coarse). Lastly, more ablation studies can make the paper stronger, e.g. ablating the temporal transfer (using just a single frame), predicting contact area versus contact point, etc. Lastly, to make the paper stronger, the authors can consider adding another baseline comparison against methods that focus on contact-rich tasks from previous literature.



**Quality Of The Limitations Section:**

Limitations are addressed clearly

**Questions For Rebuttal:**

- Is it assumed that the tool is already firmly grasped at the beginning of each episode?
- Why is sweep_to_dustpan a three-step task?
- What’s the resolution of the waypoints that determine key contact frames? This resolution seems to be slightly arbitrary. How does this resolution affect model performance?
- MPPI knows object geometry and ground truth object pose, which makes generalization to novel objects much easier compared to the PerAct baseline. During test time evaluation with novel objects, what if you feed the training object geometry to MPPI? How much will the performance degrade?
- CLIP heatmaps are probably only meaningful for nouns/adjectives. For instance, in Fig 2, the first three words “wipe”, “up”, “the”, produce meaningless heat maps. Is this a problem? If the instruction contains many words that result in irrelevant heat maps, will the model performance be affected?
- Ablate temporal transformers (using only a single frame)
- Ablate contact area estimate versus contact point estimate
- Demonstrations in simulation are cheap. If more demos are used during training, can performance be further improved?
- (Potentially) try a new, more challenging task of “applying paint”, in which the exact “shape” of the past contacts matters a lot.
- (Potentially) add another baseline comparison against methods that focus on contact-rich tasks


**Robotics Focus:**

Sufficient demonstration on hardware

**Summary Of Paper:**

The paper proposes Contact-Aware and Language conditioned spatial Action MApping for contact-RIch manipulation (CALAMARI) that outperforms SOTA models for contact-rich tasks in simulation and in the real world (with zero-shot sim2real transfer). The model takes advantage of existing VLM for pretrained spatial features to ground instruction to spatial actions (also benefit sim2real transfer) and low-level controller (MPPI) to optimize motion trajectories that maintain contact while avoiding penetration.


**Summary Of Recommendation:**

Based on the review above, I recommend Weak Reject. I am open to changing my mind during the rebuttal.
_________________
Post-Rebuttal

I have changed my recommendation to Weak Accept.

---

### Official Review · Reviewer_GLce · 2023-07-23

**Confidence:** 4
**Originality:** Good
**Technical Quality:** Good
**Clarity Of Presentation:** Very Good
**Impact:** 3

**Recommendation:**

Weak Accept: I recommend accepting the paper, but will not argue for my recommendation if the majority of other reviewers have a different opinion.

**Review:**

The paper is very well-written with a clearly-explained methodology and well-devised and analysed experiments. We've identified the following strengths and weaknesses:

Strengths:
* The paper identifies and tackles a problem that is of high importance to the robot-learning community.
* The learned policies are vision-based, making them more easy to deploy in a real-world scenario, assuming a small sim2real gap.
* Due to the simplicity of the method, the sim2real gap seems to be easier to overcome when training in simulation and deploying to a physical robot. Namely, treating contact prediction as a 2D binary mask prediction problem, which can then be overlayed on top of a point cloud and given to a planner.

Weaknesses:
* The technical contribution is rather small and incremental - both the VLM model and the MPPI controller come off-the-shelf. What is more, the method heavily depends on MPPI but it is not completely clear to the reader how does the controller work exactly, specifically for cases where multiple subgoals have to be identified - the sweeping task. How does the controller find out that contact with the surface should happen in subgoal 1, where the broom touches the surface?
* The method can only be used to train for tasks that can be demonstrated in simulation first, to have access to the ground-truth contact information.
* It is worth elaborating what do the authors mean by contact-rich? Technically, the tasks demonstrated have rather simple contact dynamics. They are undoubtedly contact-related tasks but there's nothing complex about them - i.e. there's either contact or not - there's no friction or elasticity parameter estimation occurring. What if some sponges are harder than others and we should press less when wiping the dots?

**Quality Of The Limitations Section:**

Limitations are addressed clearly

**Questions For Rebuttal:**

* How easy would it be to extend the method to tasks where we should not only estimate where contact occurs but also what type of contact - i.e. how much to press/squeeze. The authors identify this in the limitations section but is unclear if there's a way to extend the method.
* How does the subgoal identification part in MPPI work? How do we decide on the number of subgoals - 20 for the dot wiping, 3 for the sweeping task, etc. Are those somehow automatically detected in a human observation?
* Using point clouds from real sensors is usually very noisy - i.e it's common for pixels from the table to be considered part of some of the objects. Was that an issue for you and is that why we do an additional heat map encoding step, to discretise the continuous heat maps?

**Robotics Focus:**

Sufficient demonstration on hardware

**Summary Of Paper:**

The authors present a method which can extract language-guided heatmaps from RGB inputs through the usage of vision-language models. The heatmaps are then converted to binary masks which encode where in the image a contact should occur between objects/surface in the scene and a tool that a robot is holding. Those masks are then fed into an MPPI controller which estimates a robot end-effector trajectory which would satisfy the required contacts. The method is shown to work in simulation but to also transfer on a real robot seamlessly (for the chosen tasks).

**Summary Of Recommendation:**

I recommend weak accept and would be happy to change to accept depending on discussion during rebuttal.

---

### Official Review · Reviewer_ceQE · 2023-07-25

**Confidence:** 4
**Originality:** Good
**Technical Quality:** Good
**Clarity Of Presentation:** Good
**Impact:** 3

**Recommendation:**

Weak Accept: I recommend accepting the paper, but will not argue for my recommendation if the majority of other reviewers have a different opinion.

**Review:**

Language-conditioned manipulation in high-contact tasks is generally underexplored; I like this paper’s approach to the problem, reasoning over a more “semantically meaningful” action space (contact masks) in order to facilitate learning from demonstration and generalization. I also like that the overall approach is generally easy to understand (build word-specific heatmaps, then learn to aggregate to predict a contact mask — as a segmentation task).

---

However, many aspects of this paper give me cause for concern. First, I am not sure how fair the comparison is between this approach and the sole baseline — PerAct. The paper argues that PerAct needs multiple RGB-D cameras to generate calibrated point clouds, while the proposed approach (CALAMARI) only requires a single viewpoint. However, for CALAMARI to work, all parts of the scene must also be visible — undercutting this weakness. As for CALAMARI outperforming PerAct, I think this is more to do with the underlying action space dimensionality than anything else — learning in voxel space is difficult because it’s huge relative to 2D image space → wouldn’t a more fair baseline here be something along the lines of CLIPort? Especially since we’re already limiting robot actions to 4-DoF (instead of 6-DoF).

Furthermore, it’s not clear how easy it is to collect supervision in the real-world for the contact patches for the various different tasks. How is this done for real-world experiments? Is this approach scalable to larger, more long-horizon tasks?

Finally, the biggest concern I have is that of generalizability of this approach. CALAMARI assumes that 1) the single camera image captures the majority of the interactable objects in a scene, 2) that a single 2D contact mask is enough to guide manipulation, and 3) that 4-DoF actions are enough to solve the tasks. It’s very hard for me to swallow all three of these assumptions given that many real-world manipulation tasks are subject to partial observability and complex motions (e.g., pouring, twisting, moving along arcs). I don’t see how the proposed approach can scale to such applications.

---
EDIT (Post-Rebuttal): The authors did a tremendous job during rebuttal, running out many of the experiments I had originally suggested, clearing up mis-conceptions; the results from the CLIPort baseline is especially meaningful to me.

Reading the rebuttals for the other reviewers, its clear how this paper is providing a novel contribution, with strong experiments that back up the original hypotheses. I will be significantly raising my score to a weak accept!

**Quality Of The Limitations Section:**

Additional details required

**Questions For Rebuttal:**

It’s not immediately clear from the paper what assumptions you’re making over the environment to facilitate action planning (via MPPI); I understand that we’re taking predict contact masks and deprojecting to get a point cloud — but do you need further information about dynamics? What about robot arm/tool masks per timestep?

The formalism in Section 3.1 is a bit misleading; your framework seems to only allow for a single language instruction per demonstration; however, the formalism seems to suggest that language can change per timestamp... is this really true?

When generating initial heatmaps, why is it useful to use CLIP to generate heatmaps for filler or non-object words? For example, a heatmap for the word “the” is really underspecified/ambiguous — this also true for individual verbs as well, without any additional preprocessing. Can you provide more intuition for what’s actually going on here? Why not just predict heatmaps for individual objects/nouns, using off-the-shelf detectors?

**Robotics Focus:**

Sufficient demonstration on hardware

**Summary Of Paper:**

This paper introduces a new approach for language-conditioned robotic manipulation by predicting “contact patches” — sequences of 2D image masks that are used as information to guide an MPPI planner. The idea here is similar in nature to prior work in language-conditioned imitation that use object-centric or voxel-based representations. By restructuring the action space to focus on more “high-level” or “semantically meaningful” features, we can use off-the-shelf pretrained models to efficiently learn new tasks.

The approach uses CLIP to produce word-level heatmaps that are then combined to predict “contact masks” given a language prompt and current (visual) state (e.g., “wipe the dots”). These masks are then deprojected through a calibrated camera, and the resulting point cloud is used to guide an MPPI planner to produce 4-DoF robot actions. The approach is validated on two simulation environments (from RLBench) and in the real world.

**Summary Of Recommendation:**

While I like some of the ideas in this paper, I currently advocate for rejection, following the weaknesses documented above. I’d love extra clarity from the authors about the assumptions behind the approach, and general applicability of CALAMARI to more complex manipulation tasks during rebuttal.

---

### Decision · Program_Chairs · 2023-08-30

**Decision:**

Accept (Poster)

**Comment:**

This works presents a new approach for language-conditioned robotic manipulation. The approach fine tunes a pre-trained vision-language model to predict contact heat-maps in simulation and then leverages MPPI to generate end-effector trajectories that connect the contact-rich patches. The approach is evaluated in simulation against SOTA baselines and performs successfully in a zero-shot sim2real transfer on a real robot platform.

The authors have been very responsive during the rebuttal period, provided clarifications and additional evaluations and comparisons.
I believe this work will be of great interest to the CoRL Community.